# Homeostasis of Mitochondrial Ca^2+^ Stores Is Critical for Signal Amplification in *Drosophila melanogaster* Olfactory Sensory Neurons

**DOI:** 10.3390/insects13030270

**Published:** 2022-03-09

**Authors:** Eric Wiesel, Sabine Kaltofen, Bill S. Hansson, Dieter Wicher

**Affiliations:** Max Planck Institute for Chemical Ecology, 07745 Jena, Germany; ewiesel@ice.mpg.de (E.W.); skaltofen@ice.mpg.de (S.K.); hansson@ice.mpg.de (B.S.H.)

**Keywords:** *Drosophila melanogaster*, olfactory sensory neuron, odorant receptor, orco, sensitization, mitochondria, permeability transition pore, calcium imaging

## Abstract

**Simple Summary:**

The evolution of flight imposed new challenges on insects when locating and identifying food sources, mates, or enemies. As an adaptation, flying insects developed a remarkably sensitive olfactory system to detect faint odor traces. This ability is linked to the olfactory receptor class of odorant receptors, which are found in insect olfactory sensory neurons. In a subgroup of these neurons, sensitivity can be further enhanced through a process called sensitization. Extracellular calcium ions, calmodulin, and protein kinase C are known to be key factors in this process. While manipulation of mitochondrial calcium im- and export has been shown to influence odor responses in general, the connection of intracellular calcium stores to sensitization has so far been only speculative. Using two pharmacological approaches, we disrupted mitochondrial calcium management in order to explore its importance to sensitization. Overall, our findings reveal that mitochondrial calcium stores are important players in the complex intracellular signaling pathways required for sensitization.

**Abstract:**

Insects detect volatile chemosignals with olfactory sensory neurons (OSNs) that express olfactory receptors. Among them, the most sensitive receptors are the odorant receptors (ORs), which form cation channels passing Ca^2+^. OSNs expressing different groups of ORs show varying optimal odor concentration ranges according to environmental needs. Certain types of OSNs, usually attuned to high odor concentrations, allow for the detection of even low signals through the process of sensitization. By increasing the sensitivity of OSNs upon repetitive subthreshold odor stimulation, *Drosophila melanogaster* can detect even faint and turbulent odor traces during flight. While the influx of extracellular Ca^2+^ has been previously shown to be a cue for sensitization, our study investigates the importance of intracellular Ca^2+^ management. Using an open antenna preparation that allows observation and pharmacological manipulation of OSNs, we performed Ca^2+^ imaging to determine the role of Ca^2+^ storage in mitochondria. By disturbing the mitochondrial resting potential and induction of the mitochondrial permeability transition pore (mPTP), we show that effective storage of Ca^2+^ in the mitochondria is vital for sensitization to occur, and release of Ca^2+^ from the mitochondria to the cytoplasm promptly abolishes sensitization. Our study shows the importance of cellular Ca^2+^ management for sensitization in an effort to better understand the underlying mechanics of OSN modulation.

## 1. Introduction

For insects, the sense of smell plays an important role in the location of food sources, mates, breeding substrates as well as to avoid predators and other dangers [1]. To locate these points of interest, winged insects need to be able to process a wide range of concentrations, from faint filaments at a larger distance to highly concentrated odor packages near the source [2,3]. Odorant receptors (ORs), one of three distinct types of olfactory receptors in insects, have likely evolved in response to greater challenges imposed by the fickle properties of airborne odor plumes [4,5,6]. 

ORs are primarily localized in olfactory sensory neurons (OSN), which are housed in hair-like sensilla, distributed on the surface of the third antennal segment [7]. ORs are heterotetramers [8], composed of a combination of specific ligand-binding subunits (OrX) and highly conserved co-receptor proteins (Orco) [9,10]. As a complex, they form non-selective cation channels, which are both ligand-gated and cyclic-nucleotide-activated, to permeate Na^+^, K^+^, and Ca^2+^ [11,12]. 

The influx of Ca^2+^ often acts as a secondary messenger by binding to proteins such as calmodulin (CaM) [13]. Channels mediating Ca^2+^ influx usually terminate their own activity by binding to CaM to close [14]. In contrast, CaM binding seems to enhance OSN activity upon odor stimulation in certain ORs [15]. Extended Ca^2+^ influx through prolonged odor stimuli leads to an adaptation of the response, which seems to be a unique feature of ORs compared to other insect receptor classes [16]. Intracellular Ca^2+^ seems to be involved in metabotropic signaling cascades too. The PIP_2_ cleavage product IP_3_ activates IP_3_ receptors (IP_3_Rs), which in turn release Ca^2+^ from the endoplasmic reticulum to allow adaptation to prolonged odor signals [17,18].

Nonetheless, to ensure continued responsiveness, it is a vital task for any cell to restore the basal Ca^2+^ level to a tolerable range after each activity. Cytoplasmic Ca^2+^ can be extruded by the Ca^2+^-ATPase (PMCA) and the sodium/calcium exchanger (CALX) or be sequestered in the mitochondria through the mitochondrial calcium uniporter (mCU) [19,20,21,22]. Ca^2+^ stored in mitochondria can be released again by the mitochondrial permeability transition pore (mPTP) and other transporters to allow Ca^2+^ homeostasis [23,24] but seems to be involved in signal processing and odor perception as well. In mammals, mobilization of mitochondrial Ca^2+^ during odor stimulation allows extension of the dynamic response range of OSNs [25]. In insects, it has been shown that mitochondrial Ca^2+^ plays an important role in shaping the OR response [22]. In vitro, studies have also shown that the release of intracellular Ca^2+^ in response to odor stimulation leads to signal amplification [26].

Signal amplification also plays an important role for insects to dynamically adjust receptor sensitivity according to changing environmental requirements. Repeated stimulation with an odor at sub-threshold concentration can increase the sensitivity during the interval between these stimuli [27]. This process, called sensitization, relies on both Orco function as well as intracellular signaling. Most important for the sensitization process are the cyclic adenosine monophosphate (cAMP) and protein kinase C (PKC). Orco can be activated by cAMP but requires to be sufficiently phosphorylated by PKC [11,28]. An Orco mutant that cannot be phosphorylated by PKC can neither be activated by cAMP [28] nor sensitize [27]. A rise of cytosolic cAMP levels, as well as activation of PKC enzymes, are likely connected to the influx of extracellular Ca^2+^ due to OR activity [28,29]. Secondarily, proper binding of CaM to Orco is also required for sensitization in outer dendrites [30].

In the present study, we investigate if not only extracellular but also intracellular Ca^2+^ can interact with these pathways and therefore contribute to sensitization. Using calcium imaging in ex vivo preparations, two pharmacological ways of disturbing mitochondrial Ca^2+^ homeostasis were used. Induction of the mPTP by auranofin as well as depolarizing the mitochondrial resting potential by CCCP both abolished the ability to sensitize in *Drosophila melanogaster* OSNs.

## 2. Materials and Methods

### 2.1. Chemicals

VUAA1 (*N*-(4-ethylphenyl)-2-((4-ethyl-5-(3-pyridinyl)-4*H*-1,2,4-triazol-3-yl)thio)acetamide) was synthesized by the group “Mass Spectrometry/Proteomics” of the Max-Planck Institute for Chemical Ecology (Jena, Germany). Ru360 (C_2_H_26_Cl_3_N_8_O_5_Ru_2_) and carbonyl cyanide m-chlorophenylhydrazone (CCCP) were provided by Calbiochem (San Diego, CA, USA; auranofin by Sigma Aldrich (Steinheim, Germany). All non-water soluble chemicals were dissolved in dimethyl sulfoxide (DMSO) to yield a stock solution. Stock solutions in DMSO were dissolved at least 1:1000 to minimize the risk of membrane permeabilization. The use of DMSO as a solvent at that concentration was shown to not have an effect on the OR response of OSNs [22,31].

### 2.2. Fly Rearing and Antennal Preparation

*D. melanogaster* flies used in the experiments were modified by means of the GAL4/UAS binary system [32] to express the Ca^2+^-sensitive dye GCaMP6f in all Orco expressing OSNs. GCaMP6f is a highly sensitive sensor protein based on a circularly permutated green fluorescing protein (cpGFP) and CaM [33] which monitors changes in the free intracellular Ca^2+^ concentration [Ca^2+^]_I_ reporting on OR activity. The flies characterized by the genotype w;UAS-GCaMP6f/Cyo;Orco-Gal4/TM6B were reared under a 12 h light: 12 h dark cycle at 25 °C on conventional agar medium. For experiments, antennae of 4–8 day old females were excised and prepared as described in [15]. In brief, flies were anesthetized on ice. Antennae were excised and fixed in a vertical position on a glass coverslip with a two-component silicone resin (Kwik-Sil, World Precision Instruments, Friedberg, Germany) and immersed in Drosophila Ringer solution (in mM: HEPES, 5; NaCl, 130; KCl, 5; MgCl_2_, 4; CaCl_2_, 2; and sucrose, 36; pH = 7.3). The funiculus was cut at half its length allowing access to the OSNs for imaging. Air bubbles were removed. Antennae were continuously perfused with Drosophila Ringer solution in a perfusion chamber (RC-27, Warner Instruments Inc., and Hamden, CT, USA) during the experiments. 

### 2.3. Calcium Imaging 

Ca^2+^ fluorescence imaging was performed using an epifluorescence microscope (Axioskop FS, Zeiss, Jena, Germany) connected to a monochromator (Polychrome V, Till Photonics, Munich, Germany) which was controlled by an imaging control unit (ICU, Till Photonics, Gräfelfing, Germany). The microscope was equipped with a water immersion objective (LUMPFL 60×W/IR/0.8; Olympus, Hamburg, Germany). Fluorescence images were acquired using a cooled CCD camera operated with TILLVision 4.5 software (TILL Photonics, Munich, Germany). For GCaMP6f, the excitation maximum is at 497 nm, and the emission maximum is at 512 nm. GCaMP6f tagged OSNs were exited with 475 nm light at 0.2 Hz frequency with an exposition time of 50 ms. The emitted light was separated by a 490 nm dichroic mirror and filtered with a 515 nm longpass filter. For each antenna, fluorescence data from three to seven regions of interest, representing the somata of OSNs, were collected. For each treatment, ROIs of seven antennae were acquired and pooled for analysis. Experiments were comprised of 180 cycles with a sampling interval of 5 s. The Ca^2+^ response magnitude was calculated as the average ΔF/F0 in percentage [15]. Background fluorescence was subtracted, and the data were normalized to the average between the 10th and 20th cycles. 

Test compounds were applied at concentrations of VUAA1 30 µM, CCCP 1 µM, auranofin 25 µM, and Ru360 5 µM. Application of VUAA1 to elicit sensitization was performed at cycles 30 (1st), 45 (2nd), 140 (3rd), and 155 (4th). CCCP, auranofin, and Ru360 were applied at cycle 70.

### 2.4. Statistical Analysis

Statistical analyses were performed using Prism 4 (Graph-Pad Software Inc., La Jolla, CA, USA). Data are given as mean ± SEM (standard error of the mean) and were analyzed using two-tailed paired or unpaired Student’s *t*-test or One-Way-ANOVA with Tukey-Kramer post hoc multiple comparison tests. 

## 3. Results

Since ORs are Ca^2+^-permeable, any receptor activation causes a Ca^2+^ influx and a transient increase in [Ca^2+^]_i_ [11]. By ectopically expressing the Ca^2+^-sensitive sensor protein GCaMP6f in all Orco expressing OSNs, an influx of Ca^2+^ is translated to an increase in fluorescence intensity. Our experiments were performed using an open antenna preparation [15] that allowed optical and pharmacological access to the OSNs, while keeping them in contact with the sensillum lymph as well as their respective support cells. The OSNs were stimulated with the synthetic Orco agonist VUAA1 in a pairwise manner in order to produce sensitization. OR sensitization was previously observed for repeated near-threshold stimulation with time spans between 10 s and 3 min [27]. Stimulation with 30 µM VUAA1 at an interval of 75 s produced reliable sensitization for the majority of recorded neurons. 

### 3.1. Modulation of Ca^2+^ Homeostasis

For the first experiment, we tested the Ca^2+^ filling state of mitochondria under resting conditions. We used CCCP, an uncoupler of mitochondrial oxidative phosphorylation, to release the mitochondrial Ca^2+^ stores passively. CCCP disrupts ATP synthesis by transporting protons across the mitochondrial inner membrane, interfering with the proton gradient and dissipating the mitochondrial membrane potential, allowing mitochondria to release accumulated Ca^2+^ and stopping further accumulation [23].

We found that resting mitochondria maintain a small base level of Ca^2+^ stores as the application of CCCP over 550 s increased the fluorescence intensity by 19.7 ± 7.9%, which indicates an increase of 7.9% (Figure 1a). Stimulating OSNs with VUAA1 before application of CCCP increased the average fluorescence intensity to 102.3 ± 10.9% showing that Ca^2+^ imported by ORs is taken up by mitochondria and subsequently released again (Figure 1b). Overall, Ca^2+^ replenishment of the mitochondria due to OR activity led to 2.3 times higher fluorescence response compared to the unfilled state (*p* < 0.001, Figure 1c,d). 

It has been shown that mitochondrial Ca^2+^ buffering impacts the response termination and the shape of the late responses in OSNs [22]. Termination of an odor response and subsequent intracellular signaling is a critical time frame for sensitization [27]. Therefore, we evaluated if the application of CCCP might impair the OSNs ability to sensitize. While OSNs showed a signal amplification of 40.5% (*p* < 0.05) in the control between the first and second VUAA1 stimulation, after CCCP application there was no statistical difference between the pair of stimulations (Figure 2b). Instead, the release of mitochondrial Ca^2+^ seemed to evoke adaptation. After subtracting the overall increased fluorescence background from the third and fourth response the signal intensity fell by 56.2 ± 10.5% for the fourth pulse. 

### 3.2. Modulation of the Mitochondrial Ca^2+^ Export and Import

Next, we assessed if activation and inactivation of specific Ca^2+^ export and import pathways can have similar effects as impairment of Ca^2+^ homeostasis. Mitochondria can form the mPTP as a channel to release large quantities of Ca^2+^ at once [34]. Activation of the mPTP using auranofin led to a continuous increase in the fluorescence baseline to 21.2 ± 4.7% compared to 0.02 ± 2.6% in the control measurement (Figure 3a,c). Similar to treatment with CCCP, auranofin application abolished the increase in fluorescence intensity of the secondary VUAA1 pulse (Figure 3c). After subtraction of the background, the intensity of the secondary pulse dropped by 50 ± 14.6%, mirroring the CCCP results.

Mitochondria import Ca^2+^ primarily through the mCU and mCU activator and blockers have been found to influence OR responses [22,35]. We used the ruthenium amine complex Ru360 to block mitochondrial Ca^2+^ uptake [36]. Treatment with Ru360 did not affect the fluorescence base level nor the ability to sensitize (Figure 3e,f). However, the amplitude of the fluorescence response decreased from 41.5 ± 8.6% to 24.1 ± 7.4% (*p* < 0.05). 

Further, it has been shown that the blocking of mitochondrial Ca^2+^ import can rescue the effects of auranofin on OR response [22]. Simultaneous treatment with Ru360 and auranofin restored the ability to sensitize. Still, the base fluorescence increased to 12.5 ± 3%, indicating that there was Ca^2+^ stored in mitochondria before the application of Ru360 that could be released by the mPTP (Figure 3g,h).

Using the sensitization index as described in [30] to assess our data, we found that there were no quantitative differences between the effects evoked by Ca^2+^ release through CCCP or auranofin. Furthermore, the application of Ru360 did not affect the sensitization index and was able to rescue effects evoked by auranofin (Figure 4).

## 4. Discussion

Organisms constantly need to adjust the sensitivity of their sensory systems according to the signal strength of stimuli from the environment. While multiple neural circuits have been identified that mediate signal modulation and processing in *D. melanogaster* [37,38,39,40], these phenomena are not exclusive to higher brain centers. OSNs are the very first unit to receive input from odor signals and can modulate their activity through the process of sensitization. Sensitization is an example of peripheral odor information processing which is locally restricted to the sensory neurons and requires no input from the brain. As the first step in odor processing, it may be an important mechanism to adapt to varying odor concentrations. However, downstream processing in projection neurons and the antennal lobe is certainly still important for the fly’s final decision-making.

Possible pathways involved in establishing sensitization have been reviewed recently [2,41]. In short, an odor stimulus too weak to produce a robust OR activation can increase cAMP production through either transient influx of small loads of Ca^2+^ or OR dependent signaling [29]. In turn, cAMP activates Orco, which causes further Ca^2+^ influx. This influx activates two additional feedback loops. Orco activity can be further enhanced by Ca^2+^-activated CaM [15,30] and by Ca^2+^-dependent activation of PKC, which leads to phosphorylation and activation of Orco [28].

In our study, we investigated whether mitochondrial Ca^2+^ stores might play a role in these established pathways. In agreement with previous studies [19,21], we found that in *D. melanogaster* OSNs, Ca^2+^ imported due to OR activity is sequestered in mitochondria and can be released through mitochondrial transport systems (Figure 1). Release of mitochondrial Ca^2+^ stores through either dissipating the membrane potential with CCCP (Figure 2) or activation of the mPTP with auranofin (Figure 3c,d) abolished the OSN’s ability to sensitize. This effect might be caused by Ca^2+^-hotspots in the direct vicinity of ORs due to efflux from the mitochondria. Channels transferring calcium ions are known to affect the termination of their own function [14]. However, such a mechanism is yet to be described for OR channels.

Alternatively, mitochondrial stores might be directly involved in the internal signaling responsible for sensitization. Mukunda and colleagues speculated that the signaling pathway involving CaM might be driven by the release of intracellular Ca^2+^ since the effect of CaM inhibition differed qualitatively between different OrX proteins [30]. While Ru360 restored the OSNs’ ability to sensitize, the physiological response to VUAA1 was not restored in all respects. OSNs treated with auranofin and Ru360 recovered more slowly from VUAA1 stimulation than what was found in controls, with an increase in fluorescence intensity overlapping between two pulses of the sensitization (Figure 3g). Furthermore, OSNs treated with both compounds showed a reduced sensitization index of 22.6 ± 10.1% compared to 45.5 ± 10.8% of the control (*p* > 0.05). While the difference in means is not significant, other pathways of sensitization might be upregulated to offset the lack of mitochondrial Ca^2+^-signaling. Further investigations will be needed to determine the contributions of the different signaling pathways involved in sensitization. 

Previous studies have shown that sensitization might function differently in different parts of the OSNs. Modulation by CaM only appears to be significant in the outer dendritic segments [30]. Mitochondria on the other hand are exclusive to the inner dendritic segments and somata [7]. Indeed, recent studies have shown that OSN types able to sensitize, such as those housed in basiconic sensilla, have enlarged inner dendritic segments with particularly high densities of mitochondria [30,42,43]. The high density of mitochondria might be used for effective Ca^2+^ buffering allowing for very sensitive and fast Ca^2+^ signaling. 

Further, the function of mitochondria is also influenced by the endoplasmic reticulum, as they are known to be in close contact and transfer Ca^2+^ [44]. Although modulation of ER Ca^2+^ stores was previously shown to not influence OR responses [22], a possible role in sensitization cannot be ruled out and requires further investigation.

## Figures and Tables

**Figure 1 insects-13-00270-f001:**
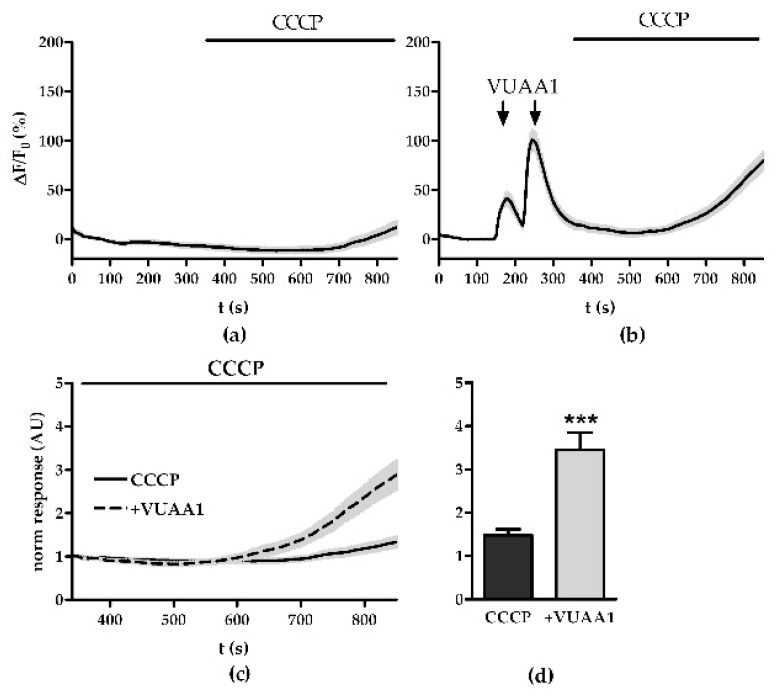
CCCP can release mitochondrial Ca^2+^ imported due to OR activity: (**a**,**b**) averaged time course of the change in Ca^2+^ fluorescence intensity ΔF/F0 in OSNs evoked by application of 1 µM CCCP under control conditions (**a**) or after a double stimulation with 30 µM VUAA1 (**b**, arrows); (**c**) comparison of normalized increase in Ca^2+^ fluorescence responses of OSNs with and without previous VUAA stimulation; (**d**) terminal Ca^2+^ responses from (**c**). Data represent mean ± SEM (CCCP, *n* = 48, CCCP + VUAA1, *n* = 49), Student’s unpaired *t*-test, *** *p* < 0.001.

**Figure 2 insects-13-00270-f002:**
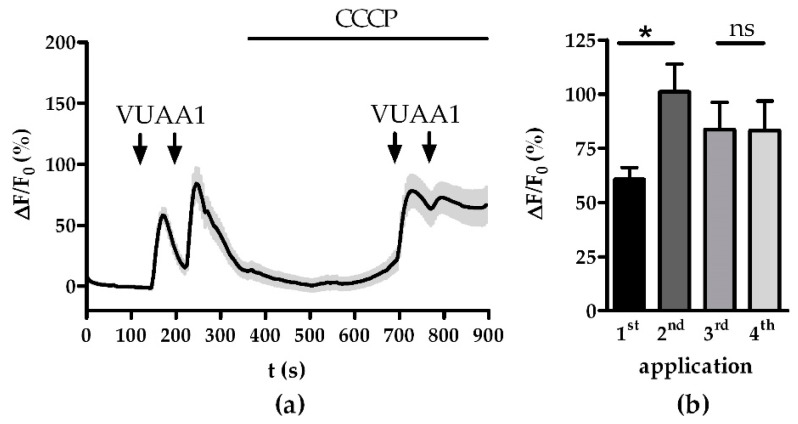
Sensitization of Orco channels is abolished by the disturbance of Ca^2+^ homeostasis by CCCP: (**a**) averaged time course of Ca^2+^ fluorescence responses to two sets of stimulations with 30 µM VUAA1 (arrows); (**b**) maximum Ca^2+^ responses for VUAA1 stimulations from (**a**). Data represent mean ± SEM (*n* = 28), Student’s paired *t*-test, ns, not significant, * *p* < 0.05.

**Figure 3 insects-13-00270-f003:**
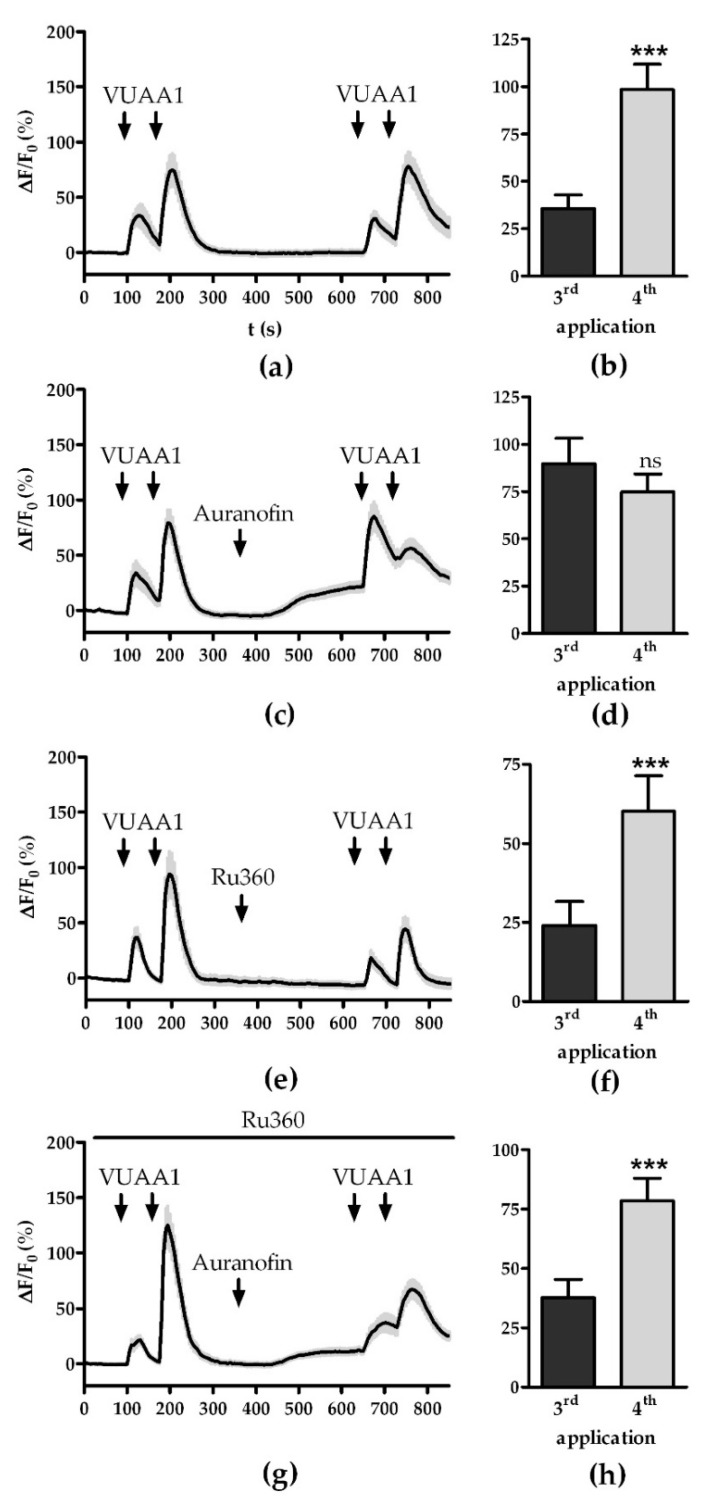
Effect of modulators of mitochondrial Ca^2+^ export and import on fluorescence time courses: (**a**,**b**) OR responses (**a**) and maximum fluorescence (**b**) upon stimulation with VUAA1 under control conditions (*n* = 31); (**c**,**d**) OR responses (**c**) and maximum fluorescence (**d**) upon stimulation with VUAA1 in absence and presence of mPTP activator auranofin (*n* = 31); (**e**,**f**) OR responses (**e**) and maximum fluorescence (**f**) upon stimulation with VUAA1 in absence and presence of mCU inhibitor Ru360 (*n* = 27); (**g**,**h**) OR responses (**g**) and maximum fluorescence (**h**) upon stimulation with VUAA1 in presence of mCU inhibitor Ru360 and mPTP activator auranofin (*n* = 26). Data represent mean ± SEM, Student’s paired *t*-test, ns *p* > 0.05, *** *p* < 0.001.

**Figure 4 insects-13-00270-f004:**
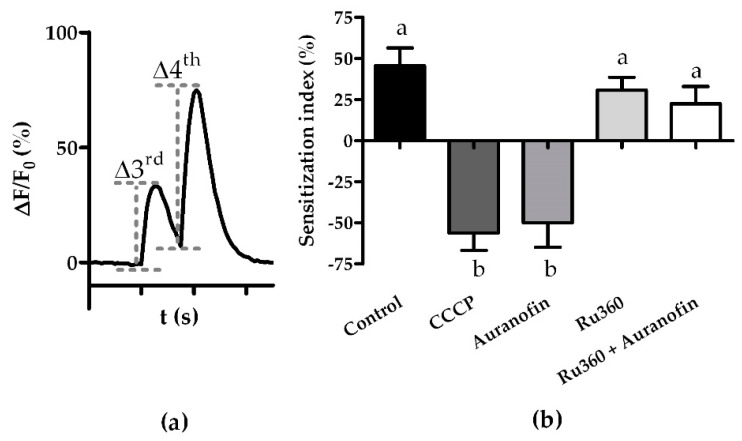
Effect of modulators of mitochondrial Ca^2+^ import and export on sensitization index: (**a**) Sensitization efficacy was quantitatively assessed by calculating the “Sensitization Index” [30]. After subtraction of the baseline fluorescence intensity before the delivery of the VUAA1 stimulation, the Sensitization Index was calculated as the difference between the intensity of the fourth and third stimulation (Δ4th-Δ3rd); (**b**) sensitization Indices after treatment with modulators of mitochondrial Ca^2+^ import and export calculated as described in (**a**). Data represent mean ± SEM, different letters above the bars indicate significant differences (*p* < 0.05) as revealed by Tukey-Kramer post hoc multiple comparison tests.

## Data Availability

The data presented in this study are available on request from the corresponding author.

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
