# Peer review of "Homeostasis of Mitochondrial Ca2+ Stores Is Critical for Signal Amplification in Drosophila melanogaster Olfactory Sensory Neurons"

_insects, 2022, doi:10.3390/insects13030270_

Round 1

Reviewer 1 Report

The authors claim that using extremely sophisticated experimental techniques, they have shown that Ca2+ stored in mitochondria is involved in the sensitization of OSNs.

To me, there is a minor question.

In their calcium imaging experiments, are the authors confirming whether the ORs that compose heteromer with Orco have the same or similar neural activity in response to the true ligand as the stimulation of Orco with VUAA1?

The authors said that Orco takes up Ca2+ by input of odor molecules that are insufficient to activate the ORs (Lines 239-244), but if that causes sensitization, wouldn't there be a relationship between mitochondrial stores of Ca2+ and amplification of sensitivity? To me, it feels like the brain is processing the generation of frequent faint signal.

Line 123
Is OLYMPUS a German company?

Author Response

Thank you very much for critical reading our manuscript and providing helpful comments.

In their calcium imaging experiments, are the authors confirming whether the ORs that compose heteromer with Orco have the same or similar neural activity in response to the true ligand as the stimulation of Orco with VUAA1?

The calcium responses produced by VUAA1 have a similar shape as these elicited by odors and sensitization has been shown to be achieved by either (e.g. Jain et al., Scientific Reports 11 (2021)3747).

The authors said that Orco takes up Ca2+ by input of odor molecules that are insufficient to activate the ORs (Lines 239-244), but if that causes sensitization, wouldn't there be a relationship between mitochondrial stores of Ca2+ and amplification of sensitivity? To me, it feels like the brain is processing the generation of frequent faint signal.

The relationship between mitochondrial stores of Ca2+ and amplification of sensitivity is indeed what we show in the manuscript. But of course, this is part of peripheral odor information processing, while further steps happen centrally.

Line 123
Is OLYMPUS a German company?

No, but we got the objective from the German branch office.

Reviewer 2 Report

Generally I think it is wery good paper. It is well-written, quite good planned and concuted the experiments. My main concern is lack of control in presented study, both untreated and after solvent treatment. If I understand correctly, authors compared the level after tratment, and before, hovewer I think that the value should be compared at the same time point in control and after treatment with compounds. Also, there is lack information how solvent itself influenced on examined parameters.

I have also few other question:

L103- Please explain why did you use mutant? 

L110 - Why femals, not femals:male?

L119 - Please explain did you conduct the same experiment on untreated and solvent treated ?

L135 - What kind of solvent did you use?

Figure 2 - Please expalain what "ns" stand for?

Disscucsion - I think that thsi section might be improve. 

Author Response

Thank you very much for critical reading our manuscript and providing helpful comments.

Authors compared the level after tratment, and before, hovewer I think that the value should be compared at the same time point in control and after treatment with compounds.

Figure 4 addresses the changes in sensitization index according to the fluorescence intensity of the second pair of stimulation (3rd and 4th stimulation). The sensitization index of the untreated control is compared to the indices of the different treatment regimes for this time point.

L103- Please explain why did you use mutant? 

The mutant fly expresses GCaMPf6, a calcium indicator reporting odorant receptor activation in the sensory neurons as these receptors are calcium permeable.

L110 - Why femals, not femals:male?

We have this standard to reduce biological variability.

L119 - Please explain did you conduct the same experiment on untreated and solvent treated ?

Solvent control has been performed and was documented in previous publications (Lucke, J., et al., Cell Calcium, 2020. 87; Prelic, S. et al., Front Cell Neurosci, 2021. 15).

L135 - What kind of solvent did you use?

The solvent was DMSO.

Figure 2 - Please expalain what "ns" stand for?

Done. N.s., not significant. Missing added to Figure 2.

Disscucsion - I think that thsi section might be improve. 

We added a statement regarding peripheral and central odor information processing.

Reviewer 3 Report

“Homeostasis of mitochondrial Ca2+ stores is critical for signal amplification in Drosophila melanogaster olfactory sensory neurons” by Wiesel et al.

In this article, the authors investigated the role of the mitochondrial Ca2+ storages in the olfactory sensitisation of Drosophila melanogaster. The experiments were well performed and the collected data were adequately described and represented. Furthermore, the conclusions are in line with what the authors has been presented and appear appropriately deepen with references.

Through calcium imaging performed in antennae preparations of mutant D. melanogaster the authors provided interesting information about the intracellular signalling pathways Ca2+-mediated in insect olfaction.

I have just a few observations:

Title: Drosophila melanogaster in italics.

Author information: please provide author affiliation

Line 145: The text should be read: Since ORs…

Lines 200-204: Please refer to the figure 3, panels e-f.

Lines 304-305 and line 247: Drosophila melanogaster can be abbreviated in D. melanogaster

Author Response

Thank you very much for critical reading our manuscript and providing helpful comments.

Title: Drosophila melanogaster in italics.

Done.

Author information: please provide author affiliation

Done.

Line 145: The text should be read: Since ORs…

Done.

Lines 200-204: Please refer to the figure 3, panels e-f.

Done.

Lines 304-305 and line 247: Drosophila melanogaster can be abbreviated in D. melanogaster

Done for l. 247, for the references we rely on the correct citation.

Round 2

Reviewer 2 Report

All my doubts have been cleared up.